# Differentiable sampling of molecular geometries with uncertainty-based adversarial attacks

Daniel Schwalbe-Koda [1,2], Aik Rui Tan [1,2] & Rafael Gómez-Bombarelli [1✉]

Neural network (NN) interatomic potentials provide fast prediction of potential energy surfaces, closely matching the accuracy of the electronic structure methods used to produce the training data. However, NN predictions are only reliable within well-learned training domains, and show volatile behavior when extrapolating. Uncertainty quantification methods can flag atomic configurations for which prediction confidence is low, but arriving at such uncertain regions requires expensive sampling of the NN phase space, often using atomistic simulations. Here, we exploit automatic differentiation to drive atomistic systems towards high-likelihood, high-uncertainty configurations without the need for molecular dynamics simulations. By performing adversarial attacks on an uncertainty metric, informative geometries that expand the training domain of NNs are sampled. When combined with an active learning loop, this approach bootstraps and improves NN potentials while decreasing the number of calls to the ground truth method. This efficiency is demonstrated on sampling of kinetic barriers, collective variables in molecules, and supramolecular chemistry in zeolite-molecule interactions, and can be extended to any NN potential architecture and materials system.

[1] Department of Materials Science and Engineering, Massachusetts Institute of Technology, Cambridge, MA, USA. [2] These authors contributed equally. Daniel Schwalbe-Koda, Aik Rui Tan. ✉email: rafagb@mit.edu

Recent advances in machine learning (ML) techniques have enabled the study of increasingly larger and more complex materials systems[1–3]. In particular, ML-based atomistic simulations have demonstrated predictions of potential energy surfaces (PESes) with accuracy comparable to ab initio simulations while being orders of magnitude faster[4–6]. ML potentials employing kernels or Gaussian processes have been widely used for fitting PESes[7–9], and are particularly effective in low-data regimes. For systems with greater diversity in chemical composition and structures, such as molecular conformations or reactions, larger training datasets are typically needed. Neural networks (NNs) can fit interatomic potentials to extensive datasets with high accuracy and lower training and inference costs[10,11]. Over the last years, several models have combined different representations and NN architectures to predict PESes with increasing accuracy[11–14]. They have been applied to predict molecular systems[15,16], solids[17], interfaces[18], chemical reactions[19,20], kinetic events[21], phase transitions[22], and many more[4,6].

Despite their remarkable capacity to interpolate between data points, NNs are known to perform poorly outside of their training domain[23,24] and may fail catastrophically for rare events, such as those occurring in atomistic simulations with large sizes or time scales not explored in the training data. Increasing the size and diversity of the training data is often beneficial to improve performance[20,25], but there are significant costs associated to generating new ground-truth data points. Continuously acquiring more data and re-training the NN along a simulation may negate some of the acceleration provided by ML models. In addition, exhaustive exploration or data augmentation of the input space is intractable. Therefore, assessing the trustworthiness of NN predictions and systematically improving them is fundamental for deploying ML-accelerated tools to real world applications, including the prediction of materials properties.

Quantifying model uncertainty then becomes key, since it allows distinguishing new inputs that are likely to be informative (and worth labeling with ab initio simulations) from those close to configurations already represented in the training data. In this context, epistemic uncertainty—the model uncertainty arising from the lack of appropriate training data—is much more relevant to ML potentials than the aleatoric uncertainty, which arises from noise in the training data. Whereas ML-based interatomic potentials are becoming increasingly popular, uncertainty quantification applied to atomistic simulations is at earlier stages[26,27]. ML potentials based on Gaussian processes are Bayesian in nature, and thus benefit from an intrinsic error quantification scheme, which has been applied to train ML potentials on-the-fly[9,28] or to accelerate nudged elastic band (NEB) calculations[29]. NNs do not typically handle uncertainty natively and it is common to use approaches that provide distributions of predictions to quantify epistemic uncertainty. Strategies such as Bayesian NNs[30], Monte Carlo dropout[31], or NN committees[32–34] allow estimating the model uncertainty by building a set of related models and comparing their predictions for a given input. In particular, NN committee force fields have been used to control simulations[35], to inform sampling strategies[36] and to calibrate error bars for computed properties[37].

Even when uncertainty estimates are available to distinguish informative from uninformative inputs, ML potentials rely on atomistic simulations to generate new trial configurations and bootstrapping a potential becomes an infinite regress problem: the training data for the potential needs to represent the full PES, but thoroughly sampling the PES requires exhaustive sampling, which needs long simulations with a stable accurate potential. It is common to perform molecular dynamics (MD) simulations with NN-based models to expand their training set in an active learning (AL) loop[20,25,38]. MD simulations explore the phase space based on the thermodynamic probability of the PES. Thus, in the best case, ML-accelerated MD simulations produce atomic configurations highly correlated to the training set that only provide incremental improvement to the potentials. Configurations corresponding to rare events may be be missing, because observing them requires large time- and size-scales that are only accessible in the final production runs after AL. In the worst case, MD trajectories can be unstable when executed with an NN potential and sample unrealistic events that are irrelevant to the true PES, especially in early stages of the AL cycle when the NN training set is not representative of the overall configuration space. Gathering data from ab initio MD prevents the latter issue, but at a higher computational cost. Some works avoid performing dynamic simulations, but still require forward exploration of the PES to find new training points[39]. Even NN simulations need to sample very large amounts of low uncertainty phase space before stumbling upon uncertain regions. Hence, one of the major bottlenecks for scaling up NN potentials is minimizing their extrapolation errors until they achieve self-sufficiency to perform atomistic simulations within the full phase space they will be used in, including handling rare events. Inverting the problem of exploring the configuration space with NN potentials would allow for a more efficient sampling of transition states and dynamic control[40,41].

In this work, we propose an inverse sampling strategy for NN-based atomistic simulations by performing gradient-based optimization of a differentiable, likelihood-weighted uncertainty metric. Building on the concept of adversarial attacks from the ML literature[42,43], new molecular conformations are sampled by backpropagating atomic displacements to find local optima that maximize the uncertainty of an NN committee while balancing thermodynamic likelihood. These new configurations are then evaluated using atomistic simulations (e.g., density functional theory or force fields) and used to retrain the NNs in an AL loop. The technique is able to bootstrap training data for NN potentials starting from few configurations, improve their extrapolation power, and efficiently explore the configuration space. The approach is demonstrated in several atomistic systems, including finding unknown local minima in a toy PES, improving kinetic barrier predictions for nitrogen inversion, increasing the stability of MD simulations in molecular systems, sampling of collective variables in alanine dipeptide, and predicting supramolecular interactions in zeolite-molecule systems. This work provides a new method to explore potential energy landscapes without the need for brute-force ab initio MD simulations to propose trial configurations.

## Results and discussion

**Theory.** An NN potential is a hypothesis function $h_\theta$ that predicts a real value of energy $\hat{E} = h_\theta(X)$ for a given atomistic configuration $X$ as input. $X$ is generally described by $n$ atoms with atomic numbers $\mathbf{Z} \in \mathbb{Z}_+^n$ and nuclear coordinates $\mathbf{R} \in \mathbb{R}^{n \times 3}$. Energy-conserving atomic forces $F_{ij}$ on atom $i$ and Cartesian coordinate $j$ are obtained by differentiating the output energy with respect to the atomic coordinates $r_{ij}$,

$$\hat{F}_{ij} = -\frac{\partial \hat{E}}{\partial r_{ij}}. \quad (1)$$

The parameters $\theta$ are trained to minimize the expected loss $\mathcal{L}$ given the distribution of ground truth data $(X, E, \mathbf{F})$ according to the dataset $\mathcal{D}$,

$$\min_\theta \mathop{\mathbb{E}}_{(X,E,\mathbf{F}) \sim \mathcal{D}} [\mathcal{L}(X, E, \mathbf{F}; \theta)]. \quad (2)$$

During training, the loss $\mathcal{L}$ is usually computed by taking the average mean squared error of the predicted and target properties

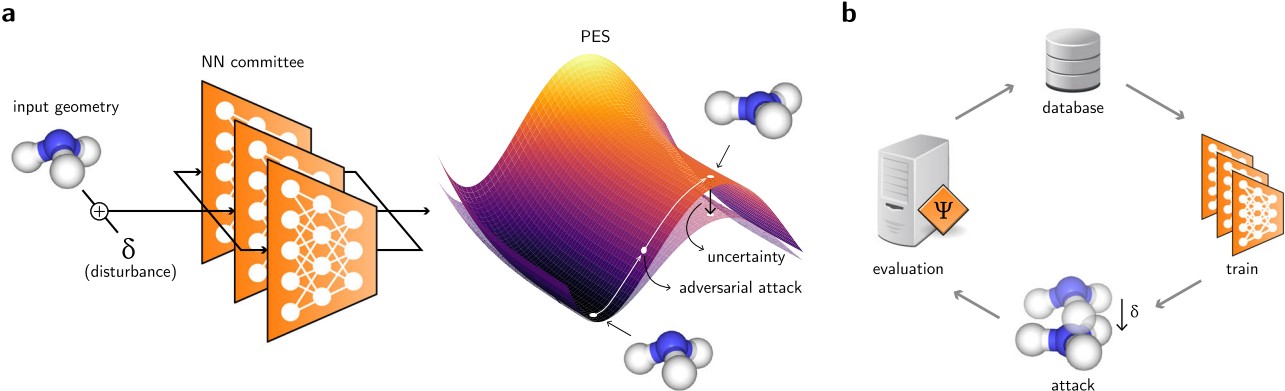

**Fig. 1 Schematic diagram of the method. a** Nuclear coordinates of an input molecule are slightly displaced by δ. Then, a potential energy surface (PES) and its associated uncertainty are calculated with an NN potential committee. By backpropagating an adversarial loss through the NN committee, the displacement δ can be updated using gradient ascent techniques until the adversarial loss is maximized, thus converging to states that compromise high uncertainty with low energy. **b** Schematic diagram of the active learning loop used to train the NN potential committee. The evaluation can be performed with classical force fields or electronic structure methods.

within a batch of size $N$,

$$\mathcal{L} = \frac{1}{N} \sum_{i=1}^{N} \left[ \alpha_E \parallel E_i - \hat{E}_i \parallel^2 + \alpha_F \parallel \mathbf{F}_i - \hat{\mathbf{F}}_i \parallel^2 \right], \qquad (3)$$

where $\alpha_E$ and $\alpha_F$ are coefficients indicating the trade-off between energy and force-matching during training[12]. The training proceeds using stochastic gradient descent-based techniques.

To create a differentiable metric of uncertainty, we turned to NN committees. These are typically implemented by training different $h_\theta$ and obtaining a distribution of predictions for each input $X$. For example, given $M$ models implementing $\hat{E}^{(m)} = h_\theta^{(m)}(X)$, the mean and the variance of the energy of an NN potential ensemble can be computed as

$$\bar{E}(X) = \frac{1}{M} \sum_{m=1}^{M} \hat{E}^{(m)}(X), \qquad (4)$$

$$\sigma_E^2(X) = \frac{1}{M-1} \sum_{m=1}^{M} \parallel \hat{E}^{(m)}(X) - \bar{E}(X) \parallel^2, \qquad (5)$$

and similarly for forces,

$$\bar{\mathbf{F}}(X) = \frac{1}{M} \sum_{m=1}^{M} \hat{\mathbf{F}}^{(m)}(X), \qquad (6)$$

$$\sigma_F^2(X) = \frac{1}{M-1} \sum_{m=1}^{M} \left[ \frac{1}{3n} \sum_{i,j} \parallel \hat{F}_{ij}^{(m)}(X) - \bar{F}_{ij}(X) \parallel^2 \right]. \qquad (7)$$

Whereas the training objective (2) rewards approaching mean energies or forces to their ground truth values, this is not guaranteed for regions outside of the training set.

Since variances in properties may become higher when the NN models are in the extrapolation regime, identifying whether an NN committee is outside its fitting domain requires evaluating the probability that the output of the NN is reliable for an input $X$. One option is to model this problem for the epistemic error as a simple classifier,

$$P(X \in \mathcal{D} \mid \sigma^2) = \begin{cases} 1, & \sigma^2 < t, \\ 0, & \sigma^2 \geq t, \end{cases} \qquad (8)$$

with $t$ a threshold chosen by evaluating the model on the training set. Although Eq. (8) can be modified to accept the data $X$ with a certain likelihood, the deterministic classifier demonstrates reasonable accuracy (see Supplementary Fig. 1 for details).

When developing adversarially robust models, the objective (2) is often changed to include a perturbation $\delta$[44],

$$\min_{\theta} \mathbb{E}_{(X,E,\mathbf{F}) \sim \mathcal{D}} \left[ \max_{\delta \in \Delta} \mathcal{L}(X_\delta, E_\delta, \mathbf{F}_\delta; \theta) \right], \qquad (9)$$

with $\Delta$ the set of allowed perturbations and $X_\delta, E_\delta, \mathbf{F}_\delta$ the perturbed geometries and their corresponding energies and forces, respectively. In the context of NN classifiers, $\Delta$ is often chosen as the set of $\ell_p$-bounded perturbations for a given $\varepsilon$, $\Delta = \{\delta \in \mathbb{R} \|\| \delta \|_p \leq \varepsilon\}$. Adversarial examples are then constructed by keeping the target class constant under the application of the adversarial attack[42,43]. On the other hand, adversarial examples are not well defined for NN regressors. Since even slight variations of the input lead to different ground truth results $E_\delta, \mathbf{F}_\delta$, creating adversarially robust NN regressors is not straightforward.

We propose that creating adversarially robust NN potentials can be achieved by combining adversarial attacks, uncertainty quantification, and active learning. Although similar strategies have been used in classifiers, graph-structured data[45,46], and physical models[47], no work has yet connected these strategies to sample multidimensional potential energy landscapes. In this framework, an adversarial attack maximizes the uncertainty in the property under prediction (Fig. 1a). Then, ground-truth properties are generated for the adversarial example. This could correspond to obtaining energies and forces for a given conformation with density functional theory (DFT) or force field approaches. After acquiring new data points, the NN committee is retrained. New rounds of sampling can be performed until the test error is sufficiently low or the phase space is explored to a desirable degree. Figure 1b illustrates this loop.

Within this pipeline, new geometries are sampled by performing an adversarial attack that maximizes an adversarial loss such as

$$\max_{\delta \in \Delta} \mathcal{L}_{\text{adv}}(X, \delta; \theta) = \max_{\delta \in \Delta} \sigma_F^2(X_\delta). \qquad (10)$$

In force-matching NN potentials, the uncertainty of the force may be a better descriptor of epistemic error than uncertainty in energy[48] (see Supplementary Figs. 1, 3–5, and 7).

In the context of atomistic simulations, the perturbation $\delta$ is applied only to the nuclear coordinates, $X_\delta = (\mathbf{Z}, \mathbf{R} + \boldsymbol{\delta})$, $\boldsymbol{\delta} \in \mathbb{R}^{n \times 3}$. For systems better described by collective variables (CVs) $\mathbf{s} = \mathbf{s}(\mathbf{R})$, an adversarial attack can be applied directly to these CVs, $X_\delta = (\mathbf{Z}, \mathbf{s}^{-1}(\mathbf{s} + \boldsymbol{\delta}))$, as long as there is at least one

differentiable function $\mathbf{s}^{-1}$ backmapping $\mathbf{s}$ to the nuclear coordinates $\mathbf{R}$.

The set $\Delta$ can be defined by appropriately choosing $\varepsilon$, the maximum $p$-norm of $\delta$. However, in atomistic simulations, it is often interesting to express these limits in terms of the energy of the states to be sampled, and the sampling temperature. To that end, a normalization constant $Q$ of the system at a given temperature $T$ can be constructed from the ground truth data $\mathcal{D}$,

$$Q = \sum_{(X,E,\mathbf{F}) \in \mathcal{D}} \exp\left(-\frac{E}{kT}\right), \qquad (11)$$

with $k$ being the Boltzmann constant. Although the form of $Q$ is inspired in the partition function of the system, it does not represent the true partition function due to the lack of data on all the states the system can exist. Accessing as many of them as possible is the required exhaustive sampling that is reserved to the production simulation after AL. Nevertheless, we can estimate that the probability $p$ that a state $X_\delta$ with predicted energy $\bar{E}(X_\delta)$ will be sampled is proportional to

$$p(X_\delta) = \frac{1}{Q} \exp\left(-\frac{\bar{E}(X_\delta)}{kT}\right). \qquad (12)$$

In this case, the factor $Q$ improves the numerical stability of the method by keeping $p(X_\delta)$ within reasonable bounds. Finally, instead of limiting the norm of $\delta$, the adversarial objective can be modified to limit the energy of sampled states by combining Eqs. (10) and (12),

$$\max_\delta \mathcal{L}_{\mathrm{adv}}(X, \delta; \theta) = \max_\delta p(X_\delta) \sigma_F^2(X_\delta). \qquad (13)$$

Using automatic differentiation strategies, the value of each displacement $\delta$ can be obtained by iteratively using gradient ascent techniques,

$$\delta^{(i+1)} = \delta^{(i)} + \alpha_\delta \frac{\partial \mathcal{L}_{\mathrm{adv}}}{\partial \delta}, \qquad (14)$$

with $i$ the number of the iteration and $\alpha_\delta$ the learning rate for the adversarial attack.

In practice, adversarial examples require input geometries as seeds, and an appropriate initialization of the displacement matrix $\delta$. One possibility is to sample the initial $\delta$ from a normal distribution $\mathcal{N}(0, \sigma_\delta^2 \mathbf{I})$ with a small value of $\sigma_\delta^2$. The degenerate case $\sigma_\delta^2 = 0$ leads to deterministic adversarial attacks with the optimization procedure.

Since one can parallelize the creation of several adversarial examples per initial seed by using batching techniques, the computational bottleneck becomes evaluating them to create more ground truth data. Hence, reducing the number of adversarial examples is of practical consideration. Generated examples can be reduced by using only a subset of the initial dataset $\mathcal{D}$ as seeds. Even then, the optimization of $\delta$ may lead to structures which are very similar, corresponding to the same points in the configuration space. To avoid evaluating the same geometry multiple times, structures can be deduplicated according to the root mean square deviation (RMSD) between the conformers. One efficient algorithm is to perform hierarchical clustering on the data points given the RMSD matrix, and aggregating points which are within a given threshold of each other. Finally, to avoid local minima around the training set, one can classify whether the given structure is well-known by the model using Eq. (8). Then, new points are evaluated only if they correspond to high uncertainty structures and not just to local optima in uncertainty, avoiding sampling regions of the PES which are already well represented in the training set.

The complete adversarial training procedure is described in Fig. 2.

---

**Algorithm:** adversarial training of an NN potential

**Input:** initial dataset $\mathcal{D}_1$, hyperparameters
**Data:** geometries, energies, and forces $(X, E, \mathbf{F})$

**for** generation $g \in (1, ..., G)$ **do**:
 // Training
 **for** model $m \in (1, ..., M)$ **do**:
 sample a train set from $\mathcal{D}_g$
 train $h_g^{(m)}$ to minimize $\mathcal{L}$ // Eqs. (2), (3)

 // Adversarial Attack
 create $Q$ from $\mathcal{D}_g$ // Eq. (11)
 sample attack seeds $\{X_i\}$ from $\mathcal{D}_g$
 **for** seed $s \in \{X_i\}$ **do**:
 initialize $\delta \sim \mathcal{N}(0, \sigma_\delta^2 \mathbf{I})$
 train $\delta$ to maximize $\mathcal{L}_{\mathrm{adv}}$ // Eqs. (13), (14)
 $X_{\delta,i} := (\mathbf{Z}_s, \mathbf{R}_s + \delta)$

 // Evaluating
 initialize $\mathcal{D}_{g+1} := \mathcal{D}_g$
 **for** adversarial example $X_{\delta,i} \in \{X_\delta\}_i$ **do**:
 obtain ground truth $(E_{\delta,i}, \mathbf{F}_{\delta,i})$ for $X_{\delta,i}$
 add $(X_{\delta,i}, E_{\delta,i}, \mathbf{F}_{\delta,i})$ to $\mathcal{D}_{g+1}$

---

**Fig. 2 Algorithm.** Pseudocode of the adversarial training of a neural network potential.

**Adversarial sampling on double well potential**. As a proof-of-concept, the adversarial sampling strategy is demonstrated in the two-dimensional (2D) double well potential (see Supplementary Note 1 and Supplementary Figs. 1–4 for an analysis of the 1D example). To investigate the exploration of the phase space, the initial data is placed randomly in one of the basins of the potential. Then, a committee of feedforward NNs is trained to reproduce the potential using the training data (see Methods). At first, the NN potential is unaware of the second basin, and predicts a single well potential in its first generation. As such, an MD simulation using this NN potential would be unable to reproduce the free energy surface of the true potential. Nevertheless, the region corresponding to the second basin is of high uncertainty when compared to the region where the training set is located. The adversarial loss encourages exploring the configuration space away from the original data, and adversarial samples that maximize $\mathcal{L}_{\mathrm{adv}}$ are evaluated with the ground truth potential, then added to the training set of the next generation of NN potentials. Figure 3a shows the results of the training-attacking loop for the NN potential after several generations. As the AL loop proceeds, the phase space is explored just enough to accurately reproduce the 2D double well, including the energy barrier and the shape of the basins.

To verify the effectiveness of the adversarial sampling strategy, the evolution of the models is compared with random sampling. While the former is obtained by solving Eq. (13), the latter is obtained by randomly selecting 20 different training points from the training set and sampling $\delta$ from a uniform distribution, $\delta \sim \mathcal{U}(-\sigma_\delta, \sigma_\delta)$. Although randomly sampling geometries is often not adequate in molecular simulation, adding small distortions to inputs has shown to increase the robustness of

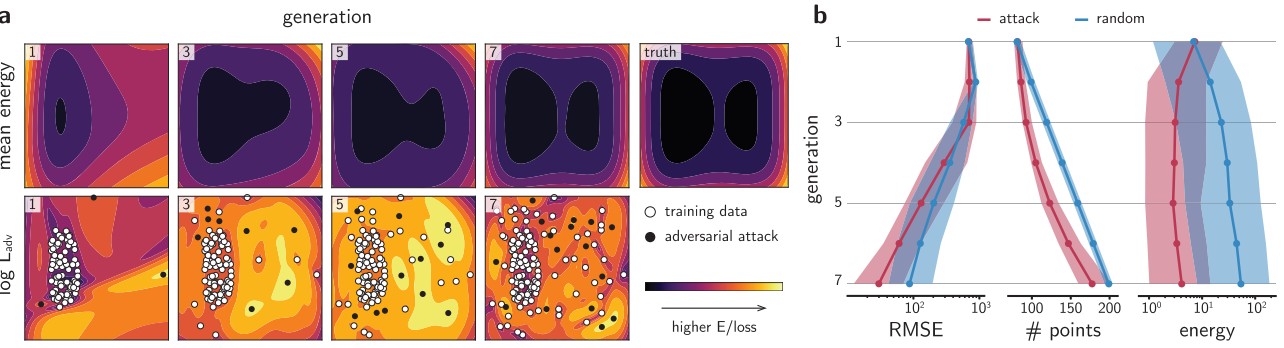

**Fig. 3 Adversarial attacks applied to a toy potential. a** Evolution of the PES of a 2D double well predicted by an NN committee. Adversarial examples (black dots) are distortions from original training data (white dots) or past adversarial examples (gray dots) that maximize the adversarial loss $\mathcal{L}_{adv}$. The plotting intervals are $[-1.5, 1.5] \times [-1.5, 1.5]$ for all graphs. The generation of the NN committee is shown in the top left corner of each graph. **b** Evolution of the root mean square error (RMSE), number of training points, and energy of the data points sampled using the adversarial attack strategy (red) or by randomly distorting the training data (blue). The solid line is the median from more than 100 experiments, and the shaded area is the interquartile region.

NN classifiers[49,50], and is a common data-acquisition technique for NN potentials, typically along vibrational normal modes[51]. Thus, it is meaningful to compare the adversarial training with random inputs for NN regressors. To perform a statistical analysis on the methods, more than 100 independent active learning loops with different initializations are trained for the same 2D well potential (Fig. 3b). Overall, the root mean square error (RMSE) between the ground truth potential and the predicted values decreases as the space is better sampled for both methods. However, although the random sampling strategy collects more data points, the median RMSE of the final generation is between two to three times higher than the adversarial attack strategy. Moreover, the median sampled energy is one order of magnitude higher for randomly sampled points. As several randomly sampled points travel to places outside of the bounds of the double well shown in Fig. 3a, the energy quickly increases, leading to high-energy configurations. This is often the case in real systems, in which randomly distorting molecules or solids rapidly lead to high-energy structures that will not be visited during production simulations. As such, this toy example suggests that the adversarial sampling method generates thermodynamically likely structures, requires less ground-truth evaluations and leads to better-trained NN potentials compared to randomly sampling the space.

**Adversarial sampling of nitrogen inversion on ammonia.** As a second example, we bootstrap an NN potential to study the nitrogen inversion in ammonia. This choice of molecule is motivated by more complex reactive systems, in which quantifying energy barriers to train a robust NN potential requires thousands of reactive trajectories from ab initio simulations[20]. To circumvent that need, we start training an NN committee using the SchNet model[12] from Hessian-displaced geometries data. Then, new geometries are sampled by performing an adversarial attack on the ground-state conformation, and later evaluated using DFT. After training a new committee with newly sampled data points, the landscape of conformations is analyzed and compared with random displacements. Figure 4a shows a UMAP visualization[52] of the conformers, as compared by their similarity using the Smooth Overlap of Atomic Positions (SOAP) representation[53]. A qualitative analysis of the UMAP plot shows that adversarial attacks rarely resemble the training set in terms of geometric similarity. Attacks from the second generation are also mostly distant from attacks in the first generation. On the other hand, small values of distortions $\sigma_\delta$ for a uniform distribution create geometries that are very similar to Hessian-displaced ones.

While higher values of $\sigma_\delta$ (e.g., $\sigma_\delta = 0.3$ Å) explore a larger conformational space, several points with very high energy are sampled (Fig. 4b), as in the double well example. As the number of atoms increases, this trade-off between thermodynamic likelihood and diversity of the randomly sampled configurations worsens in a curse-of-dimensionality effect. In contrast, energies of adversarially created conformations have a more reasonable upper bound if the uncertainty in forces is used. When the uncertainty in energy is employed in Eq. (10) instead of $\sigma_F^2$, adversarial examples may not efficiently explore the configuration space (Supplementary Fig. 5), supporting the use of $\sigma_F^2$ for performing inverse sampling. Although calculating gradients with respect to $\sigma_F^2$ requires more memory to store the computational graph (Supplementary Fig. 6), this metric is more informative of epistemic uncertainty and error in NN potentials than its energy counterpart (Supplementary Figs. 7–9) and better reflects the preference of force-matching over energy-matching at train time. Figure 4c compares the degree of distortion of the geometries with respect to their energies. It further shows that the adversarial strategy navigates the conformational space to find highly distorted, lower energy states. Both the first and second generation of attacked geometries display higher RMSD than Hessian-displaced structures with respect to the ground-state geometry while staying within reasonable energy bounds. However, as the low-energy region of the PES is better explored by the NN potential as the AL loop progresses, adversarially sampled geometries from later generations become increasingly higher in energy (Supplementary Fig. 10).

Once new configurations are used in training, predictions for the energy barrier in the nitrogen inversion improve substantially (Fig. 4d). While the first generation of the NN potential underestimates the energy barrier by about 1 kcal/mol with respect to the DFT value, the prediction from the second generation is already within the error bar, with less than 0.25 kcal/mol of error for the inversion barrier (see Supplementary Fig. 11). In contrast, predictions from an NN committee trained on randomly sampled geometries overestimate this energy. They also exhibit higher uncertainties, even for geometries close to equilibrium (Supplementary Fig. 11). This suggests that adversarial attacks were able to sample geometries that improved the interpolation of the energy barrier without the need to manually add this reaction path into the training set.

The evolution of the phase space of each NN committee is further compared in the projected PES of Fig. 4e (see Methods). Two CVs are defined to simplify the representation of the 12-

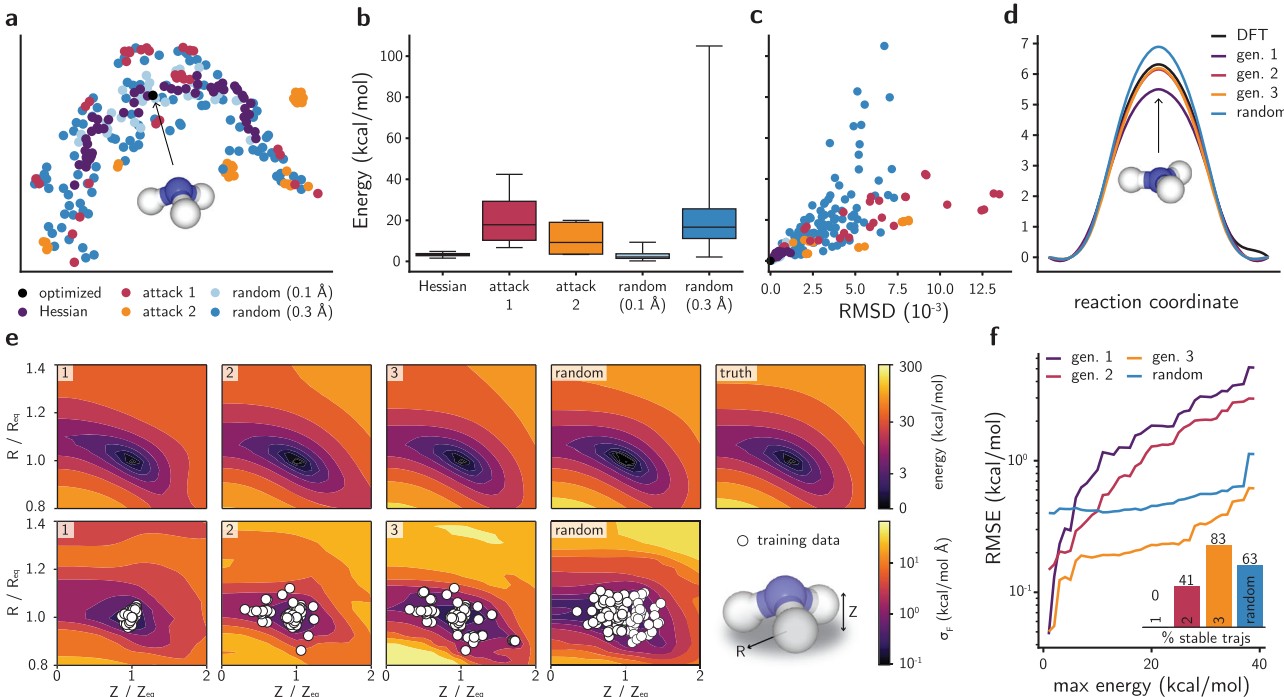

**Fig. 4 Adversarial attacks applied to the ammonia molecule. a** UMAP plot for the SOAP-based similarity between ammonia geometries. Both axes are on the same scale. **b** Distribution of DFT energies for conformations sampled with different methods. The horizontal line is the median, the box is the interquartile region and the whiskers span the range of the distribution. **c** Relationship between DFT energy and root mean square deviation (RMSD) of a geometry with respect to the ground state structure of ammonia. The color scheme follows the legend of **a**. **d** Energy barrier for the nitrogen inversion calculated with NEB using DFT or using the NN committee. **e** Evolution of the PES projected onto the CVs $(Z, R)$ for ammonia. The generation of the NN committee is shown in the top left corner of each plot. The scale bar of energies is plotted with the function $\log_{10}(1 + E)$, and all energy contour plots have the same levels. Random geometries were generated with $\sigma_\delta = 0.3$ Å (see Methods). **f** RMSE between the NN and DFT PES for each NN potential when a maximum energy is imposed for the DFT PES. **f** inset fraction of stable MD trajectories generated using each NN committee as force field.

dimensional phase space of this molecule: the radius of the circumference defined by the three hydrogen atoms $(R)$ and the distance between the nitrogen atom and the plane defined by the three hydrogens $(Z)$ (Supplementary Fig. 12). Figure 4e shows the energies and force uncertainties calculated for the most symmetrical structures containing these CVs (see Supplementary Fig. 12a), with $R, Z$ normalized by the values found in the ground state geometry. Analogously to Fig. 3a, adversarial attacks expand the configuration space used as train set for NN committees and bring the phase space closer to the ground truth, thus lowering the uncertainty of forces in the phase space (see also Supplementary Figs. 7 and 8). Nevertheless, randomly sampled geometries also allow bootstrapping an NN committee depending on the system and values of $\sigma_\delta$. Importantly, NN committees successively trained on adversarial attacks have smaller errors in the low-energy region of the PES of ammonia. As expected, the high-energy configurations sampled by randomly generated geometries slightly improve the higher energy region of the PES that will not be visited in production simulations. Figure 4f shows the RMSE of each model compared to DFT across all the projected phase space of Fig. 4e. When only energies smaller than 5 kcal/mol are compared, all three generations display much smaller RMSE than NNs trained with randomly sampled geometries, probably due to the presence of Hessian-displaced geometries in their training set. Up to 40 kcal/mol, the third generation of NN committees has a smaller RMSE when compared to committees trained with randomly distorted geometries, further supporting that the adversarial sampling strategy is useful to balance exploration of diverse conformations with higher likelihood. Finally, the adversarial training yields

models capable of performing stable MD simulations. Whereas the first generation cannot produce stable MD trajectories, i.e., always leading to unphysical configurations such as atomic dissociation or collision, 83% of the trajectories produced by the third generation of adversarially based NN committees are stable, even though the NN-based MD geometries include data points originally not in the training set (Supplementary Fig. 13). In contrast, only 63% of the trajectories are stable when the NN committee trained on random geometries is used. Since the NN committees were trained on as few as 150 training points (see Methods), this indicates that the adversarial sampling strategy enhances the robustness of NN-based MD simulations by seeking points which are known to cause instabilities due to extrapolation errors, and unlikely to exist in training sets created by unbiased MD simulations (Supplementary Fig. 13).

**Collective variable sampling in alanine dipeptide**. As a third example, we illustrate the use of adversarial attacks for sampling predefined CVs. Since translation-based adversarial attacks $X_\delta = (\mathbf{Z}, \mathbf{R} + \boldsymbol{\delta})$ may not be able to capture collective dynamics of interest such as bond rotations (see full discussion in the Supplementary Note 2), we seek high-uncertainty conformations in predefined CVs $\mathbf{s} = \mathbf{s}(\mathbf{R})$. To do that, there should exist a differentiable function $\mathbf{s}^{-1}$ mapping a point in the CV space to the atomic coordinates space $\mathbb{R}^{n \times 3}$. Typically, CVs aggregate information from many degrees of freedom and $\mathbf{s}(\mathbf{R})$ is not bijective. Nevertheless, in the case of adversarial attacks, it suffices to have an operation $\mathbf{s}^{-1}$ that acts on a geometry $X = (\mathbf{Z}, \mathbf{R})$ to produce the adversarial attack $X_\delta = \left(\mathbf{Z}, \mathbf{s}^{-1}(X, \boldsymbol{\delta})\right)$. Using this strategy, a

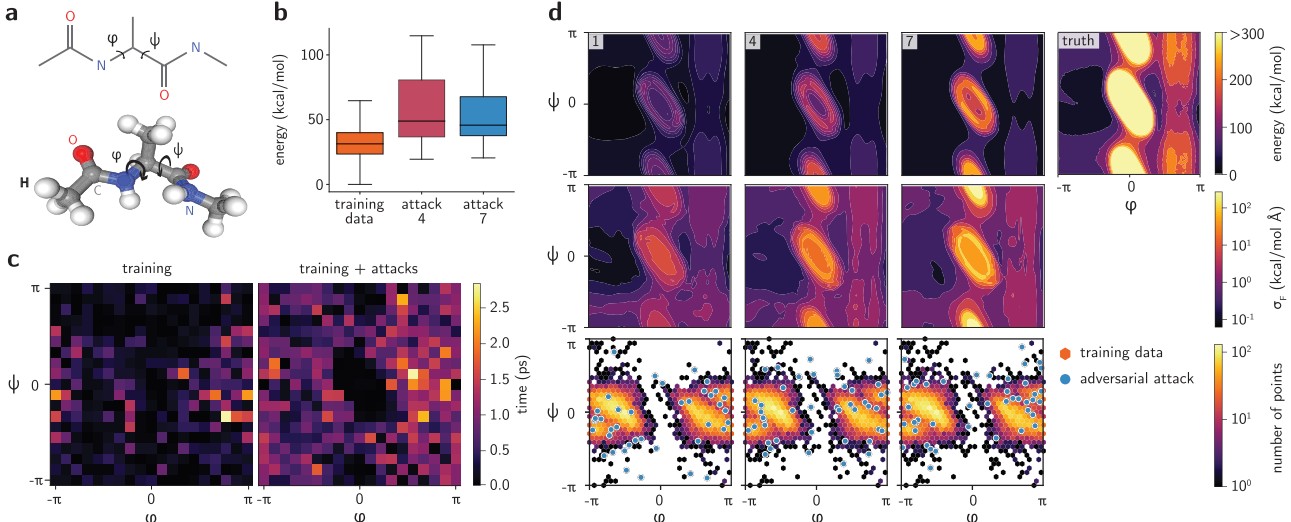

**Fig. 5 Adversarial attacks applied to an alanine dipeptide molecule. a** Schematic diagram of alanine dipeptide and the CVs ($\varphi, \psi$) created from the highlighted dihedral angles. Hydrogen, carbon, oxygen, and nitrogen atoms are depicted with white, gray, red, and blue spheres, respectively. **b** Distribution of force field energies for conformations generated from the collective variables. The horizontal line is the median, the box is the interquartile region and the whiskers span the range of the distribution. **c** Duration of stable alanine dipeptide trajectories simulated with NN committees trained with and without adversarial examples. Grid points represent dihedral angles ($\varphi, \psi$) of the starting configurations of trajectories obtained via rigid rotation of the lowest energy geometry. **d** Evolution of the PES of an NN committee trained on alanine dipeptide. Adversarial examples (blue points) are distortions along the CVs ($\varphi, \psi$) from randomly chosen original training points (hexagonal bins) through Eq. (13). Angles are given in radians.

seed geometry can be distorted in the direction of its predefined CVs even if the CVs are not invertible.

This application is illustrated with the alanine dipeptide molecule (N-acetyl-L-alanine-N′-methylamide), using its two dihedral angles ($\varphi, \psi$) as CVs (Fig. 5a). Despite their apparent chemical simplicity, flexible molecules pose tremendous challenges to NN potentials[54], which are typically benchmarked on molecules with barely any rotatable bonds (e.g., MD17). In this particular case, the function $\mathbf{s}^{-1}$ takes a reference geometry $X$ as an input and performs the dihedral rotations of interest through purely geometrical operations. Since bond rotations can be written with matrix operations, they can be implemented in the training pipeline without breaking the computational graph that enables the adversarial strategy. To compare the effects of the adversarial learning method with MD-based training sets, a series of NN committees were trained using the same architecture employed in the previous section. The models were initially trained on geometries created from MD simulations using the Optimized Potentials for Liquid Simulations (OPLS) force field[55] with the OpenMM package[56,57] (see Methods). Then, adversarial attacks were performed by randomly taking training points as seed geometries. Since bond rotations are periodic, the adversarial distortion $\delta$ does not break the geometries apart, a concern that exists in many other ML-accelerated simulations as in the previous section. Nevertheless, some angles ($\varphi, \psi$) may lead to high-energy configurations depending on the conformation $X$ of the molecule prior to the attack. Figure 5b shows the distribution of sampled energies for different rounds of adversarial attacks. We discarded points with extremely high energy from the training set, since they interfere with the training of the NN potential for being overly far from equilibrium. Nevertheless, the distribution of energies show that most of the sampled points lie in energy ranges that are not accessible by unbiased, short MD simulations, but are expected to be accessed in long production simulations. This further supports the hypothesis that adversarial attacks are effective in sampling regions of the phase space with good compromise between energy and uncertainty, even after extensive MD simulations. To confirm that the adversarial

sampling strategy improves the robustness of the NN potential, the stability of MD trajectories is computed for various initial configurations. Figure 5c compares the duration of stable trajectories obtained with the first and seventh generation of NN committees. As expected, the first generation produces many unstable trajectories, as even nanoseconds of unbiased MD simulations do not provide enough data to stabilize the NN potential. On the other hand, adding a relatively small number of adversarial examples enhances the robustness of the NN committees, as reflected in more stable MD trajectories (see also Supplementary Figs. 14 and 15). Since high-energy adversarial points are discarded from the training, the NN committee is unable to produce stable trajectories for starting configurations with CVs near ($\varphi, \psi$) = (0, 0).

The evolution of NN committees for predicting the PES of alanine dipeptide is shown in Fig. 5d. At first, only a small region of the phase space is known from the data obtained in MD simulations. This is reflected on the high contrast between the uncertainty close and far from the training set. In the first few adversarial attacks, the space is better sampled according to the uncertainty metric, decreasing the error for low-energy regions and increasing the uncertainty in high-energy regions. This suggests that the quality of the epistemic error quantification improves as the conformation space is better explored, and also further supports that epistemic error estimation is better informed by the force uncertainty (see also Supplementary Fig. 16). To better compare the ground truth results with the NN predictions in the low-energy region, we clipped the energies of the former to 300 kcal/mol in Fig. 5d. As the active learning loop progresses, the NN committee is able to better reproduce the energy landscape of alanine dipeptide, as exemplified by the improvement of the CV landscape for $\varphi > \pi/2$ or the high-energy ellipsoid centered at ($\varphi, \psi$) = (0, 0), which will not be visited in unbiased simulations. Interestingly, the uncertainty remains high in the central region, since the sampled energies of the system are much higher than the rest of the phase space. Since some of them are discarded for being extremely unlikely (e.g., configurations with energies greater than 200 kcal/mol), the predictive power of

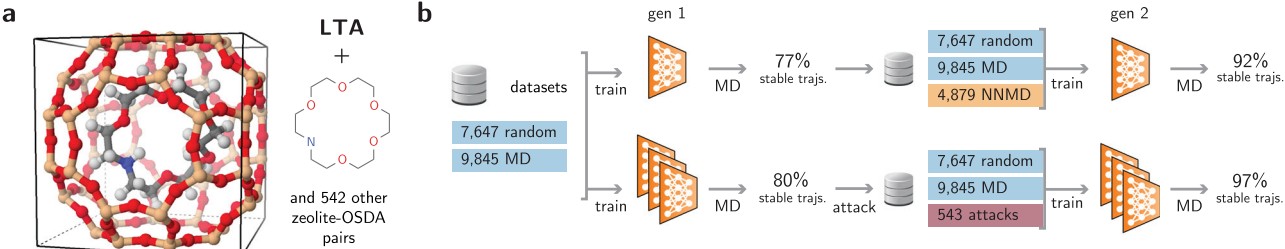

**Fig. 6 Adversarial attacks applied to zeolite-molecules systems. a** Example of zeolite-molecule pair simulated in this work. In total, 543 different poses were used as starting configurations for the simulations (see Supplementary Table 3). **b** Active learning strategies for training a NN potential for zeolite-molecule systems. A traditional, MD-based loop is compared with adversarially trained NN potentials.

the NN committee is not guaranteed in this part of the phase space. This is characterized by the ring-like energy barrier featured in Fig. 5d for later generations, and may change the absolute values of the adversarial loss (see example in Supplementary Fig. 3). It is yet unclear whether NN potentials are able to simultaneously predict ground-state conformations and such high-energy states with similar absolute accuracy[54]. In fact, learning high-energy regions of the PES may not be needed, since the learned barriers are insurmountable in production unbiased simulations. Finally, the uncertainty in forces resembles traditional biasing potentials in enhanced sampling techniques applied to obtain the free energy landscape in alanine dipeptide[58,59]. Although this intuition is not thoroughly quantified in this work, we suggest that NN potentials with uncertainty quantification intrinsically provide a bias towards transition states through the uncertainty metric. Although the uncertainty can vary outside of the training set, as seen in Fig. 5d, this idea qualitatively agrees with the examples in this paper (see also Supplementary Figs. 1a and 8). While we explore this bias through adversarial attacks for bootstrapping NN potentials in this work, we further suggest they could lead to automatic transition state and rare-event sampling strategies based on differentiable atomistic simulations with the uncertainty as a collective variable itself. The adversarial approach is compatible with other NN architectures and may be used for improving the training sets of existing models. For instance, for the ANI model[51], we have used publicly available pre-trained models and carried out uncertainty-based attacks on small molecules (see Supplementary Note 3, Supplementary Figs. 17–20). For molecules present in the training data (methane, ammonia, water), and particularly for a molecule not in the training data (alanine dipeptide), it was possible to identify high-uncertainty, thermally accessible configurations that could be added to the training data in an active learning loop.

**Adversarial attacks in solids and supramolecular chemistry**. As a final example, we show our method can be used to bootstrap NN potentials for larger systems, including solids and supramolecular chemistry. This application is illustrated with zeolite structures occluded with neutral organic molecules, some of which act as organic structure-directing agents (OSDAs) for these materials (see Fig. 6a for an example and Supplementary Table 3 for the complete list). Despite the wide commercial interest in predicting host–guest interactions in these materials, the diversity of zeolite topologies and organic molecules offers a challenge for reproducing their complex potential energy surfaces, particularly with dynamic simulations.

We start with a dataset of 543 zeolite-OSDA pairs, from which we obtain more than 17,000 DFT energies and forces (see Methods). Despite extensive data generation through MD simulations, random sampling, and structural optimizations,

NN potentials trained on this dataset are unable to fully produce stable MD trajectories. On average, 23% of the MD trajectories starting from each of the 543 optimized poses are unstable, and lead to the collapse of the simulation. This number lowers to 20% of the trajectories when an NN committee is used to perform the MD simulation (see Fig. 6b).

Conventional strategies to increase the stability of NN potentials include performing active learning loops by retraining the networks on MD-sampled data (Supplementary Fig. 21a)[20,25,36,37]. However, sampling new host–guest geometries to diversify the training of NN potentials and stabilize their predictions is computationally inefficient due to the large number of atoms in these systems. On the other hand, adversarial attacks can sample just enough new configurations to enable the models to achieve self-sufficiency in dynamic simulations. To verify this hypothesis, we performed both adversarial attacks and NN-based MD using the first generation of trained models (see Methods). In additional to the original training sets, 4879 MD frames and 543 adversarial attacks were evaluated using single-point DFT calculations and added to the training set of the next generation (Fig. 6b). After retraining the NN potentials, new NN-based MD simulations were performed. NNs trained with MD trajectories are less stable than their adversarially robust counterparts despite being trained with nine times more new points. Whereas 8% of the trajectories produced by the second generation of MD-trained NNs are unstable, only 3% of the trajectories produced with adversarially robust NNs are unphysical. Even when the possible stabilizing effect by the NN committee is disregarded, the MD-trained NNs use significantly more data points to achieve a similar performance. While the overall cost of performing DFT calculations can be lowered by filtering out geometries with low uncertainty[36,37] or using the deduplication techniques discussed in the Theory section, MD simulations may not maximally sample informative points for retraining the NNs since they are bound to overrepresent low-energy minima and scarcely visit highly informative rare events. Hence, adversarially sampled geometries enable evaluating fewer points with DFT-level calculations while improving the performance of the neural networks, showing increasing advantage in larger and more diverse systems. The method might enable NN potentials to be applied in increasingly complex and realistic materials systems.

In summary, we proposed a new sampling strategy for NN potentials by combining uncertainty quantification, automatic differentiation, adversarial attacks, and active learning. By maximizing the uncertainty of NN predictions through a differentiable metric, new geometries can be sampled efficiently and purposefully. This technique allows NN potentials to be bootstrapped with fewer calls to the ground truth method, maximizing the final accuracy and efficiently exploring the conformational space. Successful adversarial attacks were demonstrated in four examples. In a 2D double well potential, the attacks provided an exploration strategy and outperformed a

random baseline. In the ammonia molecule, the approach accurately predicted distorted configurations or reaction paths, and produced better fits to the PES and more stable atomistic simulations, without the need of AIMD. For alanine dipeptide, a challenging molecule for NN potentials due to its flexibility, adversarial attacks were performed on collective variables to efficiently explore phase space and systemically improve the quality of the learned PES. Finally, for zeolite-molecule systems, sampling new data points with adversarial attacks leads to more robust NN potentials with less training points compared to MD-based active learning loops. This work presents a new data-efficient way to train NN potentials and explore the conformational space through deep learning-enabled simulations. By balancing thermodynamic likelihood and attacking model confidence it becomes possible to gather representative training data for uncertain, extrapolative configurations corresponding to rare events that would otherwise only be visited during expensive production simulations. The approach can be extended to any NN-based potential, such as the publicly available ANI, and representation, and can be further explored for biased simulations.

## Methods

**Double well potential**. The double well potential adopted in this work is written as the following polynomial:

$$E(x, y) = 10x^4 - 10x^2 + 2x + 4y^2. \tag{15}$$

Initial training data was generated by randomly sampling up to 800 points with independent coordinates according to a uniform distribution $\mathcal{U}(-1.5, 1.5)$, and selecting only those with energy lower than $-2$. This allows us to select only data points lying in the lowest energy basin of the double well, creating an energy barrier between the two energy minima.

Five feedforward NNs with four layers, softplus activation and 1024 units per layer were trained using the same train/test splits of the dataset. The NNs had different initial weights. The dataset was split in the ratio $60:20:20$ for training : validation : testing, with a batch size of 35. The training was performed for 600 epochs with the Adam optimizer[60] and a learning rate of 0.001. The reported RMSE is the root mean squared difference between the average predicted energy $\bar{E}$ and the ground truth potential $E$ as evaluated on a $100 \times 100$ grid in the region $[-1.5, 1.5] \times [-1.5, 1.5]$.

Adversarial attacks were performed with a normalized sampling temperature of 5 (Eq. (15) units) for 600 epochs, learning rate of 0.003 and the Adam optimizer. Deduplication via hierarchical clustering was performed using a threshold of 0.02 for the distance and the 80th percentile of the train set variance.

Random distortions were performed in each generation by displacing the $(x, y)$ coordinates of training data points (or past random samples) by $\delta \sim \mathcal{U}(-1.0, 1.0)$. After deduplication via hierarchical clustering and uncertainty percentile as performed for adversarial attacks, up to 20 points were randomly selected from the resulting data. Distortions smaller than 1.0 were often unable to efficiently explore the PES of the double well, landing in the same basin.

**Simulations of ammonia**. Initial molecular conformers were generated using RDKit[61] with the MMFF94 force field[62]. DFT structural optimizations and single-point calculations were performed using the BP86-D3/def2-SVP[63,64] level of theory as implemented in ORCA[65]. NEB calculations[66,67] were performed with 11 images using the FIRE algorithm[68] as implemented in the Atomic Simulation Environment[69]. Hessian-displaced geometries were created by randomly displacing the atoms from their ground-state conformation in the direction of normal mode vectors with temperatures between 250 and 750 K. In total, 78 training geometries were used as initial dataset.

For each generation, five NNs with the SchNet architecture[12] were employed. Each model used four convolutions, 256 filters, atom basis of size 256, 32 learnable gaussians and cutoff of 5.0 Å. The models were trained on different splits of the initial dataset (ratios $60:20:20$ for train : validation : test) for 500 epochs, using the Adam optimizer with an initial learning rate of $3 \times 10^{-4}$ and batch size of 30. A scheduler reduced the learning rate by a factor of 0.5 if 30 epochs passed without improvement in the validation set. The training coefficients $\alpha_E$ and $\alpha_F$ (see Eq. (3)) were set to 0.1 and 1, respectively.

Adversarial attacks were initialized by displacing the ground-state geometry of ammonia by $\delta \sim \mathcal{N}(0, 0.01$ Å) for each coordinate. The resulting attack $\delta$ was optimized for 60 iterations using the Adam optimizer with learning rate of 0.01. The normalized temperature $kT$ was set to 20 kcal/mol to ensure that adversarial attacks were not bound by a low sampling temperature, but by the uncertainty in force predictions. 30 adversarial attacks were sampled for each generation. No deduplication was performed.

Random distortions were generated by displacing each coordinate of the ground-state geometry of ammonia by a value of $\delta \sim \mathcal{U}(-\sigma_\delta, \sigma_\delta)$. The values of $\sigma_\delta = 0.1$ Å and $\sigma_\delta = 0.3$ Å were adopted. 30 (100) random samples were created for $\sigma_\delta = 0.3$ Å ($\sigma_\delta = 1.0$ Å).

NN-based MD simulations were performed in the NVT ensemble with Nosé-Hoover dynamics, 0.5 fs timesteps, and temperatures of 500, 600, 700, 800, 900, and 1000 K. 100 5 ps-long trajectories were performed for each NN committee and temperature. The ground-state geometry of ammonia was used as initial configuration for all MD calculations. Trajectories were considered as unphysical if the distance between hydrogen atoms was closer than 0.80 Å or larger than 2.55 Å, or if the predicted energy was lower than the ground-state energy (0 kcal/mol for the reference adopted in this work).

SOAP vectors were created using the DScribe package[70]. The cutoff radius was set as 5 Å, with spherical primitive Gaussian type orbitals with standard deviation of 1 Å, basis size of 5 functions, and $L_{max} = 6$. The vectors were averaged over sites before summing the magnetic quantum numbers.

The projected PES shown in Fig. 4e is constructed by evaluating the NN potentials on symmetrical geometries generated for each tuple $(Z, R)$. As such, train points and adversarial attacks are projected onto this space even though the conformers display distortions not captured by the CVs $(Z, R)$ (see Supplementary Fig. 12). The RMSE between the projected PES of the NN potential and DFT calculations is taken with respect to these symmetrical geometries.

**Simulations of alanine dipeptide**. Alanine dipeptide was simulated using the OPLS force field[55] within the OpenMM simulation package[56,57]. The force field parameters were generated using LigParGen[71]. The molecule was placed in vacuum, with a box of size 30 Å. MD simulations were performed at 1200 K using a Langevin integrator with a friction coefficient of 1 ps$^{-1}$ and step sizes of 2 fs. Calculations of Lennard-Jones and Coulomb interactions were performed in real space with no cutoff. The initial training data was obtained by conducting 320 ns of MD simulations, from which snapshots every 2 ps were collected. 10,000 snapshots were extracted from these trajectories as the initial training data for the NN committee.

For each generation, five NNs with the SchNet architecture[12] were employed. The NNs follow the same architecture employed in the simulation of ammonia, with five NNs per committee, each containing four convolutions, 256 filters, atom basis of size 256, 32 learnable gaussians and cutoff of 5.0 Å. The models were trained on different splits of the initial dataset (ratios $60:20:20$ for train : validation: test) for 200 epochs, using the Adam optimizer with an initial learning rate of $5 \times 10^{-4}$ and batch size of 50. A scheduler reduced the learning rate by a factor of 0.5 if 30 epochs passed without improvement in the validation set. The training coefficients $\alpha_E$ and $\alpha_F$ (see Eq. (3)) were both set to 1.0.

Adversarial attacks were initialized by displacing the CVs $(\varphi, \psi)$ by $\delta \sim \mathcal{N}(0, 0.01$ rad) for each angle. The resulting attack $\delta$ was optimized for 300 iterations using the Adam optimizer with learning rate of $5 \times 10^{-3}$. Normalized temperature of $kT$ was set to 20 kcal/mol. 50 adversarial attacks were sampled for each generation. No deduplication was performed.

NN-based MD simulations were performed in the NVE ensemble using Velocity Verlet integrator at 300 K with a timestep of 0.5 fs. Trajectories starting from 324 different initial configurations were performed for each NN committee. Each starting geometry was obtained via rotation of the dihedral angles of the ground-state configuration while keeping the connected branches rigid. Trajectories were considered unstable if distance between bonded atoms became smaller than 0.75 Å or larger than 2.0 Å.

**Simulations of zeolites**. DFT calculations of zeolite-OSDA systems were performed using the Vienna Ab-initio Simulation Package (VASP)[72,73], version 5.4.4, within the projector-augmented wave (PAW) method[74,75]. The Perdew–Burke–Ernzerhof (PBE) functional within the generalized gradient approximation (GGA)[76] was used as the exchange-correlation functional. vdW interactions were taken into account through Grimme's D3 corrections[77,78]. The kinetic energy cutoff for plane waves was restricted to 520 eV. Integrations over the Brillouin zone were performed using Monkhorst-Pack $k$-point meshes[79] (Γ-centered for hexagonal unit cells) with a uniform density of 64 $k$-points/Å$^{-3}$ (see Supplementary Table 2). A threshold of $10^{-6}$ eV was adopted for the energy convergence within a self-consistent field (SCF) cycle. Relaxation of unit cell parameters and atomic positions was performed until the Hellmann–Feynman forces on atoms were smaller than 10 meV/Å.

543 different poses were created by docking 107 neutral molecules into 66 pure-silica zeolite frameworks with the VOID package[80]. Then, poses were fully optimized using DFT, following the guidelines in ref. [81]. In total, 253 different zeolite-molecule pairs with less than 350 atoms were selected for computational efficiency (see Supplementary Table 3 for complete details). Within the same zeolite-molecule complex, poses differ according to the initial placement of the molecule or loading[81].

An initial dataset of structures was created by displacing atoms of each pose by up to 0.2 Å. Then, each new structure was calculated using DFT at the PBE-D3 level. The procedure was repeated about 14 times per structure, resulting in about 7647 geometries. In addition, 9184 off-equilibrium configurations of zeolite-OSDA pairs sampled using AIMD simulations within the NVT ensemble at 600 K were

added to the initial training set. Similarly, 661 frames from unloaded, pure-silica zeolites sampled using the NVT ensemble at 450 K were added to the training set.

A conventional active learning loop was performed by retraining the NN potentials on geometries sampled with NN-based MD simulations[20,25] performed in the NVE ensemble using the Velocity Verlet integrator with initial temperature of 600 K, a duration of 5 ps, and a timestep of 0.5 fs. Trajectories started from each of the 543 optimized poses. For each trajectory, 10 frames corresponding to the last 2 ps of the simulations were saved and later evaluated using DFT. Frames with DFT energy higher than 20 kcal/mol/atom above its ground state, often due to unstable trajectories, were not added to the training sets. When evaluating the robustness of the models, NN-based MD simulations were performed using the same parameters described above, but at a higher temperature of 1000 K. Trajectories were considered unstable if distances between bonded atoms became smaller than 0.75 Å or larger than 2.0 Å throughout the simulation.

Two scenarios were considered for each generation of neural networks: (i) a single NN is retrained from its own MD simulations[20,25]; or five NNs are retrained with geometries sampled using adversarial attacks (see Supplementary Fig. 21). All NNs employ the same SchNet architecture, with five NNs per committee, each containing four convolutions, 256 filters, atom basis of size 256, 64 learnable gaussians, and cutoff of 5.0 Å. The models were trained on different splits of the initial dataset (ratios 60:20:20 for train:validation:test) for 400 epochs, using the Adam optimizer with an initial learning rate of $5 \times 10^{-4}$ and batch size of 150. A scheduler reduced the learning rate by a factor of 0.5 if 25 epochs passed without improvement in the validation set. The training coefficients $\alpha_E$ and $\alpha_F$ (see Eq. (3)) were set to 0.1 and 1.0, respectively.

Adversarial attacks were initialized by displacing the atomic coordinates of optimized geometries by $\delta \sim \mathcal{N}(0, 0.01 \text{ Å})$ for each pose. The resulting attack $\delta$ was optimized for 200 iterations using the Adam optimizer with a learning rate of $10^{-2}$. The normalized temperature $kT$ was set to 20 kcal/mol. No deduplication was performed.

## Data availability

The atomistic simulation data generated in this study have been deposited in the Materials Cloud Archive under accession code https://doi.org/10.24435/materialscloud:2w-6h[83].

## Code availability

The code used to reproduce the results from this paper is available at https://github.com/learningmatter-mit/Atomistic-Adversarial-Attacks under the MIT License (see ref.[84] for permanent link).

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

## Acknowledgements

D.S.-K. acknowledges the MIT Energy Fellowship for financial support. A.R.T. thanks Asahi Glass Company for financial support. R.G.-B. acknowledges support from ARPAe DIFFERENTIATE DE-AR0001220. The DFT calculations from this paper were executed at the Massachusetts Green High-Performance Computing Center with support from MIT Research Computing, and at the Extreme Science and Engineering Discovery Environment (XSEDE)[82] Expanse through allocation TG-DMR200068.

## Author contributions

D.S.-K. conceived the project. D.S.-K. and A.R.T. designed the experiments, performed the simulations, and wrote the computer code. R.G.-B. supervised the research. All authors contributed to the data analysis and manuscript writing.

## Competing interests

The authors declare no competing interests.
