## [Peer Review File · Nature Communications]

REVIEWER COMMENTS

Reviewer #1 (Remarks to the Author):

The authors outline a sampling approach to bootstrap training datasets for machine-learned (ML) potentials, which is adapted from adversarial attacks in ML literature. The key idea is to add new configurations to the training set for which the model can not give confident answers. To find such configurations, an initial input is perturbed to maximize an uncertainty measure. Said uncertainty is obtained as the sample variance of a model ensemble and weighted by a likelihood that limits the energy of the sampled states. The optimization can also be steered along path of collective variables to bias the sampling, e.g. bond rotations. Adversarial sampling is positioned as an alternative to composing training datasets via random sampling or MD simulations. The viability of this approach is demonstrated on three toy problems, a double-well potential and two molecules with 4 (ammonia) and ~20 atoms (alanine dipeptide).

While the proposed sampling algorithm is simple and elegant, I have doubts whether it truly offers computational advantages over existing sampling schemes. Traditionally, training datasets are generated from converged MD trajectories based on relatively inexpensive DFT calculations at a temperature that is higher than the intended ML-based simulation temperature. This ensures that all relevant areas of the PES are captured in the dataset. The authors give a rather pessimistic perspective on this process and state that it is (A) expensive and (B) leads to ML-models that are unstable in MD simulations. I will address those two points separately.

(A) While generating converged MD trajectories as a basis for the training set is certainly not cheap, it is competing with the proposed process of

- (1) Generating an initial training set,
- (2) training a statistically stable ensemble of NN models on different subsets of the reference data,
- (3) performing the optimization of the adversarial loss for multiple different seed geometries,
- (4) repeating this process for multiple generations.

Especially for the considered toy problems, I question whether this is actually cheaper. After all every new configuration is obtained as the result of an optimization problem (maximization of the uncertainty measure) involving the whole ensemble of models. The manuscript does not address this question rigorously.

(B) Extracting a training set from converged MD trajectories, as outlined above, virtually guarantees that the correct data distribution is captured. This suggests that the reported stability issues in MD simulation could be due artifacts of the employed models, rather than a sampling issue per se. In light of this, the quoted success rates of the ML-based MDs (63% for MD-based training sets, 83% after three generations of the adversarial sample procedure) seem rather low for the considered systems. This needs to be checked very carefully as there might be a bug.

Furthermore several crucial questions are left unanswered:

It seem that the uncertainty measure could grow very large in high energy regions (e.g. in the repulsive limit) and potentially overbear the likelihood weighting. The authors indicate that sometimes very high energy configurations are sampled, which seems to hint at that. Is there a robust way to deal with such a "runaway" scenario of the uncertainty maximisation procedure? How does this method avoid jumping between different PESs (corresponding to different energy levels) when sampling (and calculating) extreme configurations? Is this sampling scheme robust and efficient in very high dimensions (for systems with many more atoms)?

Overall the manuscript is easy to follow and the experiments are well-chosen to illustrate the core concept. Using adversarial sampling techniques to bootstrap PES reconstruction is a novel and non-obvious idea, which is why I think that the work ultimately represents a valuable contribution. However I feel strongly that the advantages over current sampling approaches might be oversold/should be substantiated better. As a starting point, the manuscript may benefit from addressing the issues raised above.

Reviewer #2 (Remarks to the Author):

The manuscript identifies correctly that the largest source of error in machine learned force fields is comes from the limited range of configurations in the training data set. It introduces a constructive approach for generating new training data that are identified as being low energy and also high in predicted variance (according to a committee model). The idea represents an advance over established active learning tools, which which merely act as "filters", selected configurations from a larger set that is generated in some unspecified way (usually by sampling methods) to be added to the training set.

While the idea is a good one, I see the present manuscript as more of a proof of principle, rather than demonstrating that the method actually works and can have a real impact. There are two major shortcomings:

(i) The baseline against which the constructive approach is benchmarked is "random sampling". This is a "straw man". In practice, as the introduction suggests, some relevant measure would be used to obtain potentially useful new training data, which would then be "filtered" according to some error metric. The authors should therefore benchmark their approach against this as the baseline, using the very same error metric that they develop (the variance of the committee). Moreover, the baseline sampling really cannot be uniform random sampling, as that is not realistic. In practice most people would use molecular dynamics, typically at a higher temperature than what is envisaged as the "production" situation, i.e. the temperature at which the errors are then measured.

(ii) The tests systems in the manuscript are too low dimensional. Of course it is OK to use a double well potential for explanatory and pedagogical purposes, but the ammonia inversion is also effectively a one dimensional problem (there is only one slow(er) degree of freedom which is explored constructively), and the alanine dipeptide example is also just an effectively two dimensional problem. The reason this is a problem for the manuscript is that such low dimensional systems are easy to sample with the non-constructive approaches that the manuscript is trying to improve upon. Barriers are low, and slightly elevated temperature molecular dynamics would sample all relevant parts of the landscape easily and efficiently. The real test of the method would come from higher dimensional examples: either longer polymer chains, where it takes much more effort to find new low energy conformations. Another alternative is to try it on molecular solids for example, and see whether it would be able to discover and fit multiple crystal phases.

I also have a number of smaller, more technical comments.

1. It would be interesting to see whether the variance from the committee model actually correlates with the real error of the model, especially for the larger dimensional examples. high dimensional fits are notorious for such intrinsic error measure being only "qualitative" and not quantitative. The former is fine for the normal active learning selection, but the present method relies the the committee error being a good proxy for the real error.

2. I don't quite see the relevance of the "partition function" Q introduced in the manuscript. It does not appear to be a true partition function, because it is only summed over the training set, but is then used as the denominator for new configurations that are not in the training set - so it does not appear to be the correct normalisation constant. But why is such normalising needed? the $p(x)$ factor, even unnormalised, would bias the constructed configuration towards low energies.

3. Could the present method be used for "quality assurance" of an already existing production model? E.g. the ANI force field is published along with a small committee, could the method be used to construct configurations at which ANI is particularly bad? That would be an interesting and rather useful contribution.

Massachusetts Institute of Technology
77 Massachusetts Avenue
Cambridge, MA 02139

June 14, 2021

Reviewer 1

***Reviewer 1:** The authors outline a sampling approach to bootstrap training datasets for machine-learned (ML) potentials, which is adapted from adversarial attacks in ML literature. The key idea is to add new configurations to the training set for which the model*

can not give confident answers. To find such configurations, an initial input is perturbed to maximize an uncertainty measure. Said uncertainty is obtained as the sample variance of a model ensemble and weighted by a likelihood that limits the energy of the sampled states. The optimization can also be steered along path of collective variables to bias the sampling, e.g. bond rotations. Adversarial sampling is positioned as an alternative to composing training datasets via random sampling or MD simulations. The viability of this approach is demonstrated on three toy problems, a double-well potential and two molecules with 4 (ammonia) and 20 atoms (alanine dipeptide).

While the proposed sampling algorithm is simple and elegant, I have doubts whether it truly offers computational advantages over existing sampling schemes. Traditionally, training datasets are generated from converged MD trajectories based on relatively inexpensive DFT calculations at a temperature that is higher than the intended ML-based simulation temperature. This ensures that all relevant areas of the PES are captured in the dataset. The authors give a rather pessimistic perspective on this process and state that it is (A) expensive and (B) leads to ML-models that are unstable in MD simulations. I will address those two points separately.

Authors: We thank the reviewer for the positive evaluation of our work and the comments. Below, we address the suggested improvements and clarify the points raised by the referee.

Reviewer 1: (A) While generating converged MD trajectories as a basis for the training set is certainly not cheap, it is competing with the proposed process of

- (1) Generating an initial training set,
- (2) training a statistically stable ensemble of NN models on different subsets of the reference data,
- (3) performing the optimization of the adversarial loss for multiple different seed geometries,
- (4) repeating this process for multiple generations.

Especially for the considered toy problems, I question whether this is actually cheaper. After all every new configuration is obtained as the result of an optimization problem (maximization of the uncertainty measure) involving the whole ensemble of models. The manuscript does not address this question rigorously.

Authors: Our bootstrapping strategy is designed to reduce two sources of computational cost and wall time: (a) the need for running long *serial* MD trajectories (either with NN or DFT) for sampling the rare events that may be visited in the production simulation and thus need to be captured in the training data; and (b) calling the expensive ground truth evaluation on new configurations as rarely as possible. From the reviewer’s list, we argue that (1) (2) and (4) are also needed for existing approaches, and remain either unchanged or improved with the differentiable sampling. We argue that (3) greatly outperforms the alternative of running NN-based MD.

One of the most important aspects of NN potentials is that they easily allow for batching of

samples at evaluation time, even if used as committees. Figure S6 shows the GPU memory allocated for the simulation with increasing batch sizes. When five NNs using the SchNet architecture (see Methods) are used for evaluation, up to 1,000 ammonia molecules can be allocated in 2 GB of GPU memory. Interestingly, all NNs in a committee can also be trained simultaneously in the same batches and lead to different models, as long as they have different random initializations. We have tested this approach with statistical power using the 2D double well example by performing the active learning loop over 100 times (Fig. 2b), and no difference was found between training the models separately or combined. This remarkable parallelization scheme shows that items (2) and (3) from the reviewer do not significantly increase the computational cost compared to the traditional training methods. Furthermore, many current NN-MD approaches on ensembles may stabilize trajectories or at least use their uncertainty quantification to decide which NN frames are worth revisiting with ground truth simulations, thus paying off the cost of (2).

In particular, the optimization procedure (3) requires only a few GPU-seconds to be performed for thousands of systems (tested in a single NVIDIA Quadro RTX 6000 GPU), since the backpropagation can be applied to a batch of systems. The optimization is also fast thanks to the availability of gradients, requiring less than 200 iterations to converge (see Figs. S10, S17, S19). 200 iterations of the adversarial attack is roughly equivalent to 200 fs of MD simulation in wall time. While the NN-MD trajectories are also parallelizable, the diversity of the configurational space they visit is much more bound by simulated time and may not produce as many meaningful frames in 200 fs as 200 steps of adversarial attacks.

Generating an initial training set (1) is equally an issue for any NN potential and we suggest that our bootstrapping method can minimize the need for a large set of training points given to the NN at (1), as the system will self-improve effectively. This also reduces the computational cost in (2). This leanness remains through the iterations: fewer calls to the NN are needed to find informative frameworks, fewer new DFT simulations are needed, and the training data does not get bloated with uninformative frames. In contrast with MD simulations, the adversarial sampling strategy selects points which are maximally informative for improving the NN, while MD trajectories likely include extremely redundant information in the training set, so accessing new informative configurations is a rare event, for which long production simulations would be needed. If a fully-equilibrated trajectory is needed to gather the training data, one needs to do exhaustive sampling but the potentials does not yet exist. Our new example on zeolite-molecule interactions showcases how simply adding more data from MD simulations does not necessarily improve the neural network performance, even if one order of magnitude more data points are included in the training set (see full discussion with Reviewer 2). We believe that studying how diverse or redundant should training sets be to train NN potentials can be investigated in a follow-up work.

Finally, we avoid sequential MD simulations by parallelizing the sampling of new geometries. This is particularly useful in supercomputer centers, where hundreds or thousands of calculations can be carried in parallel, both single-point DFT calculations when la-

being new frames, or the adversarial attacks, thus reducing the wall time of each active learning loop. Several groups have observed that many active learning loops are required to train accurate models, even if relying on MD simulations to generate the data [1–7]. The use of uncertainty quantification is also becoming widespread, and may be adopted as a standard for machine learning simulations. Therefore, it is possible that all training strategies for NN potentials will require both active learning loops and uncertainty quantification in the future. In this scenario, our sampling strategy significantly outperforms MD simulations in terms of computational cost.

Action taken: we clarify this point in the manuscript (lines 89–94, 105–108, 120–122, 283–285).

Reviewer 1: (B) Extracting a training set from converged MD trajectories, as outlined above, virtually guarantees that the correct data distribution is captured. This suggests that the reported stability issues in MD simulation could be due artifacts of the employed models, rather than a sampling issue per se. In light of this, the quoted success rates of the ML-based MDs (63% for MD-based training sets, 83% after three generations of the adversarial sample procedure) seem rather low for the considered systems. This needs to be checked very carefully as there might be a bug.

Authors: We agree with the reviewer that the instability of NN-MD trajectories is not an issue with the sampling strategy, but with the NN potentials. The machine learning community is dedicating a lot of effort to understand the vulnerabilities of NNs to bring them to production purposes [8–12]. Even in classification problems, where there is a discrete number of classes to predict, NNs show the vulnerabilities that led to the discovery of adversarial attacks. In NN regressors, these vulnerabilities might be exacerbated by when the NNs are in the extrapolation regime. Very recent work highlights the current difficulties in fitting flexible molecules and rare events in graph-convolutional NN potentials [13], which we chose to highlight with the alanine dipeptide example.

In this context, the question addressed in the paper is not only about sampling configurational space of molecules and materials in general, but also how to sample geometries that will improve the robustness of neural network potentials. It is known that NNs display vulnerabilities due to their high dimensionality [8, 9], and preventing these problems is an active area of research in the machine learning community. It is through this targeted sampling that we address the artifact mentioned by the reviewer.

Our study of the ammonia system shows how such bootstrapping can lead to accurate systems even at the low-data regime. Only about 140 data points in total were used to train the NN potential at the third generation. A similar number of data points for randomly-sampled geometries were used for comparison. At this small-data regime, NNs are less robust and may be more prone to extrapolation errors, which makes it more interesting to explore. In contrast, the ANI-1x dataset¹ has 3600 data points for the

¹Retrieved from <https://github.com/aiqm/torchani>

ammonia molecule.

In our manuscript, we show that simply adding more data points to the training data, even if diverse in terms of energy and distortion (see Figs. 2a-c), does not necessarily improve the NN. If the goal was to simply maximize the stability of the NNs without the data constraint, we agree with the reviewer that an MD simulation would be appropriate. However, since we also tackle this problem from the perspective of improving the robustness of NNs with small amounts of data, we constrain the sampling of new data points. Our results for ammonia show that adding less than 30 well-selected points per loop significantly improves the stability of the trajectories, increasing from 0 to 41% from the first to the second generation, and 41% to 83% from the second to the third generation of NNs. In contrast, 63% of trajectories from NNs trained with randomly sampled geometries were stable. To avoid the confusion we gather from the reviewer’s comment, we changed the label on the inset of Fig. 3f from “`rnd`” to “`random`”.

These advantages are even more clear in our new example of zeolite-molecules. Whereas creating a large training set using MD is feasible for a few systems, as mentioned by the referee, performing the same level of simulations for numerous topologies and occluded molecules is overly expensive. Our new example highlights that adversarial samples may improve the robustness of NN force fields without the need for many more data points.

Finally, very few works address the issues of stability and extrapolation power in NN potentials, even in the machine learning community. We believe our work can bridge the worlds of adversarial attacks and atomistic simulations and spur research to improve extrapolation capabilities of NN potentials for molecules and materials.

Action taken: this point was clarified in the manuscript (lines 489–491).

***Reviewer 1:** Furthermore several crucial questions are left unanswered:*

It seem that the uncertainty measure could grow very large in high energy regions (e.g. in the repulsive limit) and potentially overbear the likelihood weighting. The authors indicate that sometimes very high energy configurations are sampled, which seems to hint at that. Is there a robust way to deal with such a “runaway” scenario of the uncertainty maximisation procedure?

Authors: In our manuscript, we exemplified the reviewer’s observation with the case of alanine dipeptide. In this system, high-energy configurations were sampled in the repulsive region of the PES. However, they were not sampled because of uncertainty dominating the energy term, which is unlikely since the weighting factor decreases exponentially with increasing energies and likely rises faster than the non-linearity in the NN. What happens, in fact, is that the magnitude of *predicted* energy decreases much faster than it should in the ground truth PES, because NN potentials will revert to low-energy, relatively flat PES predictions for unseen configurations. This artificially increases the value of $p(X_\delta)$ (see Fig. S2a for a simple example with the 1D double well potential). Fig. 4d shows that the region around $(\varphi, \psi) = (0, 0)$ has indeed lower predicted energy than its surroundings,

which is not accurate and is an example of the "return to the mean" described above. We believe that NN potentials might be unable to simultaneously fit high- and low-energy regions with comparable accuracy. This inability of graph-convolutional NN potentials to represent different regions of the PES with equal accuracy should be discussed in more depth [13], but is still an open-ended question.

This implies that our biasing strategy improves as the accuracy of the NN potentials increases. In many cases, sampling high-energy states might be desirable to prevent further generations of NNs from exploring that particular area of the PES. This is what is seen with the double well examples. On the other hand, our exploration strategy can be constrained by changing the equivalent temperature kT used in the weighting scheme to prevent extremely high energies from being explored. For the case of alanine dipeptide, we used a high temperature value, with $kT = 20$ kcal/mol, but a lower value would constrain the exploration of the configuration space to lower energies. In the answer below, we continue this discussion about sampling different regions of the PES.

Action taken: this explanation was added to the manuscript (lines 605–607).

Reviewer 1: How does this method avoid jumping between different PESs (corresponding to different energy levels) when sampling (and calculating) extreme configurations?

Authors: Jumping between different regions of the PES may or may not be desirable depending on the applications. The double well examples show a case in which crossing barriers and finding new states is desirable. On the other hand, the alanine dipeptide case shows that climbing a barrier has led to an undersampled, repulsive energy region that the NN potential is not able to correctly learn. In our method, the degree to which the configuration space is explored through adversarial attacks depends on the equivalent temperature kT in Eqs. (11) and (12) of the main paper. Figures S2c,d show the effect of modulating this temperature on the adversarial loss. At low temperatures ($kT = 0.3$ in Fig. S2d), the adversarial loss does not go beyond the double well minima and the configuration space is not explored beyond the existing data, as would also happen in traditional MD simulations. At intermediate values of temperature ($kT = 3$ in Fig. S2d), the adversarial loss rewards exploration of the barrier and a few points beyond the energy minima of the double well potential, but quickly drops as the energy increases for $|r| > 1.7$. Finally, at very high temperatures ($kT = 30$ in Fig. S2d), the adversarial attacks explore the configuration space even further from the training data, since exploration is not bound by the energy anymore. Therefore, if the user is interested in sampling a single PES, as suggested by the reviewer, using a lower temperature may be desirable. If the NN potential is expected to operate in higher energies, such as reactive conditions, exploring with a higher normalization temperature can be interesting. In general, because the differentiable sampling uses a gradient-based local optimization technique, it will typically drift away from a given region of PES, and only escape it if it shows very low uncertainty.

Reviewer 1: Is this sampling scheme robust and efficient in very high dimensions (for

systems with many more atoms)?

Authors: Yes, the method works well for much larger systems. Here, we decided to include an additional example of adversarial sampling of host-guest interactions in zeolites (see complete discussion with Reviewer 2). We show that this much more complex system, with diverse structures, compositions, and topologies, can be described using NN potentials improved using the adversarial sampling strategy.

Reviewer 1: Overall the manuscript is easy to follow and the experiments are well-chosen to illustrate the core concept. Using adversarial sampling techniques to bootstrap PES reconstruction is a novel and non-obvious idea, which is why I think that the work ultimately represents a valuable contribution. However I feel strongly that the advantages over current sampling approaches might be oversold/should be substantiated better. As a starting point, the manuscript may benefit from addressing the issues raised above.

Authors: Once again, we thank the reviewer for the insights. We hope this letter and the changes made to the manuscript further clarify the main points of the paper.

Reviewer 2

Reviewer 2: The manuscript identifies correctly that the largest source of error in machine learned force fields is comes from the limited range of configurations in the training data set. It introduces a constructive approach for generating new training data that are identified as being low energy and also high in predicted variance (according to a committee model). The idea represents an advance over established active learning tools, which which merely act as “filters”, selected configurations from a larger set that is generated in some unspecified way (usually by sampling methods) to be added to the training set. While the idea is a good one, I see the present manuscript as more of a proof of principle, rather than demonstrating that the method actually works and can have a real impact.

Authors: We thank the reviewer for the positive evaluation of our work and the comments. Below, we address the suggested improvements, clarify the points raised by the referee, and describe a new application that exemplifies the use of our method in a production case.

Reviewer 2: There are two major shortcomings: (i) The baseline against which the constructive approach is benchmarked is “random sampling”. This is a “straw man”. In practice, as the introduction suggests, some relevant measure would be used to obtain potentially useful new training data, which would then be “filtered” according to some error metric. The authors should therefore benchmark their approach against this as the baseline, using the very same error metric that they develop (the variance of the

committee). Moreover, the baseline sampling really cannot be uniform random sampling, as that is not realistic. In practice most people would use molecular dynamics, typically at a higher temperature than what is envisaged as the “production” situation, i.e. the temperature at which the errors are then measured.

Authors: We understand the reviewer’s concern with using random sampling as a baseline. It is important to note that for the more realistic, higher-dimensional problems of alanine dipeptide and the new examples with organic molecules in nanoporous silicates, we mostly use unbiased MD as the baseline, except for the very first generation.

In the lower dimensional examples, we use random sampling as a baseline. Although this choice can seem unnatural for the materials/chemistry communities, the machine learning community discovered that adding random noise to images at train time led to more robust NN classifiers [14–18]. It is also a common data-acquisition technique for NN potentials, typically along vibrational normal modes [22]. From the perspective of machine learning, therefore, the random sampling baseline should be the first one to consider when investigating the robustness of NNs. If a more robust NN could be obtained simply by exploring the configuration space using random displacements, this would not justify the use of a more complex sampling technique. The adversarial training strategy injects distortions tailored to the models at train time to increase the robustness of NNs [11]. Although these concepts have been studied in depth for image classification, few examples are reported for NN regressors. In our paper, we have shown how adversarial training improves the robustness of NN regressors when compared to random sampling both for a toy example and an atomistic system.

In our third example on the alanine dipeptide, we do exactly what the reviewer suggested (see Methods section). The training data is first created by performing MD simulations at a high temperature, and then used to train a NN potential. From the stability of the systems, we show that this approach is still not enough to generate accurate, stable NN potentials (see also the discussion with Reviewer 1). On the other hand, by adding as few as **50** extra points sampled using adversarial attacks per generation, we were able to significantly increase the stability of these NN potentials (see Methods, Fig. 4c, and Fig. S14). Moreover, we have found that unless a significant number of data points from MD simulations are added to the training data, the stability of NNs does not necessarily improve (see Fig. 5b). Instabilities in NN potentials often happen not because of a whole region of the PES was not sampled, but because one particular distortion might lead to extremely wrong forces (e.g., Fig. S12), which get integrated and move the system further away from physically meaningful configurations. This leads to the collapse of the simulation. Our method is convenient for finding these points with precision, instead of brute-forcing through the configuration space through MD simulations to find these particular geometries.

To comply with the reviewer’s comments, however, we included a new application of the method on training a NN force field for zeolites, where we benchmark the method against conventional strategies in active learning (see full discussion below).

Our method, therefore, is different from the typical approaches used in the machine learning community, and uses insights from materials simulations to improve NN potentials.

Action taken: added Section III.D, V.D, Fig. 5, and comment to lines 342–349.

Reviewer 2: (ii) The tests systems in the manuscript are too low dimensional. Of course it is OK to use a double well potential for explanatory and pedagogical purposes, but the ammonia inversion is also effectively a one dimensional problem (there is only one slow(er) degree of freedom which is explored constructively), and the alanine dipeptide example is also just an effectively two dimensional problem. The reason this is a problem for the manuscript is that such low dimensional systems are easy to sample with the non-constructive approaches that the manuscript is trying to improve upon. Barriers are low, and slightly elevated temperature molecular dynamics would sample all relevant parts of the landscape easily and efficiently. The real test of the method would come from higher dimensional examples: either longer polymer chains, where it takes much more effort to find new low energy conformations. Another alternative is to try it on molecular solids for example, and see whether it would be able to discover and fit multiple crystal phases.

Authors: Compressing the high-dimensional molecular systems into low-dimensional representations may suggest our examples are simplified models. However, the case of alanine dipeptide, for example, is much more complex than a two dimensional system, even though we focus our results on the two dihedral angles to exemplify the usefulness of adversarial attacks into collective variables. In fact, while the dihedral rotations are slow degrees of freedom, other faster bond rotations and vibrations pose a challenge for a system with high chemical diversity such as alanine dipeptide. The improvements shown by our NNFF strategy shows that dealing with these flexible molecules is more difficult than typically discussed in the literature. Other groups have been only recently began reporting many limitations of neural networks when dealing with rotational degrees of freedom [13], confirming that dealing with these systems is not as simple as it seems.

We agree that automatically sampling very high-dimensional systems would be a remarkable achievement. Other works have shown different strategies to sample systems such as proteins [19]. Nevertheless, developing this computational infrastructure for systems such as the ones suggested by the reviewers are beyond the scope of this manuscript. We believe that this work goes beyond merely proof-of-concept, and that this is a transferable strategy that will enable us and others to achieve these goals of differentiable enhanced sampling across a wide range of materials, including the ones suggested by the referee.

To enrich the discussion towards another class of materials and provide further evidence of the usefulness of our method, as suggested by the reviewer, we used our strategy to fit a NN potential to predict host-guest dynamics in zeolites. Zeolites are widely used as industrial catalysts and separators, but force fields are still unable to accurately predict some relevant properties of these materials, including framework flexibility, templating ability, or catalytic activity. The simulated 66 different zeolite topologies and 107 different molecules occluded in these materials, exemplifying that the systems are extremely high-

dimensional and cannot be interpreted in terms of a few collective variables.

We started the analysis by fitting a NN potential to data acquired using *ab initio* MD simulations for zeolites with charge-neutral OSDAs [20, 21]. Despite following the traditional strategies to generate the data and using tens of thousands of data points, the resulting NN potentials still generate unstable MD trajectories at levels inadequate for production purposes. By using our sampling strategy on top of these models, we were able to retrain the NN potentials, significantly improving their stability. As a result, we were able to obtain stable dynamics of organic structure-directing agents in these frameworks. This suggests our method can be used as a stepping stone to study more advanced effects in these systems, including catalysis, synthesis, and diffusion. These topics can be explored in detail in follow-up works.

We believe this additional example demonstrates that our method is not only generalizable to arbitrary systems, but has significant implications for materials simulations. Initial poses studied in this work will be available in the GitHub repository of this manuscript, along with some pre-trained models.

Action taken: Added a complete discussion on: Sections III-D, V-D, Fig. 5, Fig. S21, and Tables S2-3.

Reviewer 2: I also have a number of smaller, more technical comments.

1. It would be interesting to see whether the variance from the committee model actually correlates with the real error of the model, especially for the larger dimensional examples. high dimensional fits are notorious for such intrinsic error measure being only "qualitative" and not quantitative. The former is fine for the normal active learning selection, but the present method relies the the committee error being a good proxy for the real error.

Authors: We had already shown this analysis for the 1D double well potential (see Fig. S1). As the reviewer correctly points out, the variance correlates qualitatively to the error. While a high uncertainty does not imply a high error, points with high error typically show high uncertainty. This relationship is enough for our method to work correctly. Even if the differences between the true and estimated values are small, a high variance suggests that the region of the potential energy surface is undersampled, since different NNs predict different values of energy and forces. Figure S3 is a didactic example of that case, particularly after generation 4. We can see that even when the error of the barrier is small, the uncertainty is high because the region does not have enough training data. This automatic exploration strategy, even when relying on a qualitative metric, is the major difference between our method and other sampling strategies.

In addition to this existing analysis for the 1D double well, we added the correlation between error and uncertainty for the ammonia and alanine dipeptide systems. The results are similar to the discussion above, with high uncertainty being a good predictor of either undersampled regions or geometries with high error. More importantly, points with high error always have high uncertainty, suggesting that the neural networks do

not predict wrong values with high confidence, as often happens in NN classifiers under adversarial attacks.

Action taken: Added Figs. S9 and S16.

Reviewer 2: 2. I don't quite see the relevance of the "partition function" Q introduced in the manuscript. It does not appear to be a true partition function, because it is only summed over the training set, but is then used as the denominator for new configurations that are not in the training set - so it does not appear to be the correct normalisation constant. But why is such normalising needed? the $p(x)$ factor, even unnormalised, would bias the constructed configuration towards low energies.

Authors: Indeed, Q is not the true partition function for the system. Unfortunately, while training/attacking the NNs, we do not have access to all states of the system in order to construct the actual partition function, since we operate with limited data. Nevertheless, we borrow the idea provided by this normalization constant for two reasons: (i) making the biasing potential agnostic to the reference level chosen by the user, and (ii) controlling the absolute values of the uncertainty gradients during attacks. We explain both cases below.

For (i), the reference energy for the training data is arbitrary. As a consequence, the exponential term of Eq. (12) of the paper,

$$p(X_\delta) = \frac{1}{Q} \exp\left(-\frac{\bar{E}(X_\delta)}{kT}\right),$$

could become arbitrarily small or large if Q was not included, and would depend on the reference chosen by the user. This is undesirable from the perspective of numerical stability. Although the referee is correct in saying that the exponential term would bias the loss function following the same trend as without the factor Q , there may be numerical errors due to overflow or truncation if the exponential factor is too small or large.

Factor (ii) is a consequence of this rescaling. Gradients of the adversarial loss may become too small or large depending on the values of energy, destabilizing the adversarial training of the neural network. This could be prevented by simply changing the learning rate α_δ of the adversarial attack (see Eq. (14) of the paper). However, modifying this learning rate for each system would require some trial and error, making our method slightly harder to be used in practice. Therefore, we normalization constant Q plays an important role in making this approach practical even when, in principle, the results could be the same without it.

Action taken: We replaced the term "partition function" to "normalization constant" in the manuscript. Although we still indicate that Q is inspired in a partition function, we also clarify the lack of data in the main text (lines 253, 257–268)

Reviewer 2: 3. Could the present method be used for "quality assurance" of an already existing production model? E.g. the ANI force field is published along with a small com-

mittee, could the method be used to construct configurations at which ANI is particularly bad? That would be an interesting and rather useful contribution.

Authors: Our method is agnostic to the NN architecture, but relies on differentiable simulations to sample new geometries. As long as the map between energy and atomic coordinates can be computed with gradients, say through an autodifferentiation package, the adversarial sampling strategy should work. The ANI force field uses Behler-Parrinello symmetry functions to represent atomic environments before feeding them to NN committees [22], and it is possible to write this representation using such differentiable codes.

To investigate the question proposed by the reviewer, we used the TorchANI code and the models released with it [23]. By adapting our the code to perform adversarial attacks using the ANI architecture, we were able to sample new geometries outside of the training set of ANI. Figs. S17-S20 compare the RMSD, geometries, forces and energies of adversarially sampled geometries and the training set of ANI-1x. We found that the adversarial strategy is able to produce molecular geometries for which predictions of the ANI-1x models are not as accurate as those within the training set. Despite the small values of RMSD compared to the training set, the adversarial sampling strategy obtains systems for which the uncertainty in forces is on the order of 10 kcal/mol/Å. In addition to creating more robust NN potentials, we believe our approach may improve the generation of datasets containing off-equilibrium geometries for training NNs.

Action taken: we added this discussion to the Supplementary Text (Section I.C of the Supplementary Materials), along with Figs. S17-S20 and Table S1. A line pointing to this was added to the main text (lines 629–640). The code to reproduce the use of adversarial attacks to the ANI-1x force field was added to the GitHub repository of this work.

*References

- [1] Garrido Torres, J. A., Jennings, P. C., Hansen, M. H., Boes, J. R. & Bligaard, T. Low-Scaling Algorithm for Nudged Elastic Band Calculations Using a Surrogate Machine Learning Model. *Physical Review Letters* **122**, 156001 (2019).
- [2] Jinnouchi, R., Lahnsteiner, J., Karsai, F., Kresse, G. & Bokdam, M. Phase Transitions of Hybrid Perovskites Simulated by Machine-Learning Force Fields Trained on the Fly with Bayesian Inference. *Physical Review Letters* **122**, 225701 (2019).
- [3] Vandermause, J. *et al.* On-the-fly active learning of interpretable Bayesian force fields for atomistic rare events. *npj Computational Materials* **6**, 20 (2020).
- [4] Wang, W., Yang, T., Harris, W. H. & Gómez-Bombarelli, R. Active learning and neural network potentials accelerate molecular screening of ether-based solvate ionic liquids. *Chemical Communications* **56**, 8920–8923 (2020).
- [5] Ang, S. J., Wang, W., Schwalbe-Koda, D., Axelrod, S. & Gómez-Bombarelli, R. Active learning accelerates ab initio molecular dynamics on reactive energy surfaces. *Chem* 1–32 (2021).
- [6] Schran, C., Brezina, K. & Marsalek, O. Committee neural network potentials control generalization errors and enable active learning. *Journal of Chemical Physics* **153**, 104105 (2020).
- [7] Imbalzano, G. *et al.* Uncertainty estimation for molecular dynamics and sampling. *The Journal of Chemical Physics* **154**, 74102 (2021).
- [8] Szegedy, C. *et al.* Intriguing properties of neural networks. *2nd International Conference on Learning Representations, ICLR 2014 - Conference Track Proceedings* 1–10 (2014).
- [9] Goodfellow, I. J., Shlens, J. & Szegedy, C. Explaining and harnessing adversarial examples. *3rd International Conference on Learning Representations, ICLR 2015 - Conference Track Proceedings* 1–11 (2015).
- [10] Barrett, D., Hill, F., Santoro, A., Morcos, A. & Lillicrap, T. Measuring abstract reasoning in neural networks. In Dy, J. & Krause, A. (eds.) *Proceedings of the 35th International Conference on Machine Learning*, vol. 80 of *Proceedings of Machine Learning Research*, 511–520 (PMLR, Stockholmsmässan, Stockholm Sweden, 2018).
- [11] Tsipras, D. *et al.* Robustness May Be at Odds with Accuracy. In *International Conference on Learning Representations*, 161–168 (2018).
- [12] Xu, K. *et al.* How neural networks extrapolate: From feedforward to graph neural networks. *arXiv:2009.11848* (2020).
- [13] Vassilev-Galindo, V., Fonseca, G., Poltavsky, I. & Tkatchenko, A. Challenges for machine learning force fields in reproducing potential energy surfaces of flexible molecules. *The Journal of Chemical Physics* **154**, 94119 (2021).
- [14] Li, B., Chen, C., Wang, W. & Carin, L. Certified Adversarial Robustness with

- Additive Noise. *arXiv:1809.03113* (2018).
- [15] Gilmer, J., Ford, N., Carlini, N. & Cubuk, E. Adversarial Examples Are a Natural Consequence of Test Error in Noise. In Chaudhuri, K. & Salakhutdinov, R. (eds.) *Proceedings of the 36th International Conference on Machine Learning*, vol. 97 of *Proceedings of Machine Learning Research*, 2280–2289 (PMLR, 2019).
- [16] Lecuyer, M., Atlidakis, V., Geambasu, R., Hsu, D. & Jana, S. Certified Robustness to Adversarial Examples with Differential Privacy. *arXiv:1802.03471* (2018).
- [17] Phan, H. *et al.* Scalable Differential Privacy with Certified Robustness in Adversarial Learning. In III, H. D. & Singh, A. (eds.) *Proceedings of the 37th International Conference on Machine Learning*, vol. 119 of *Proceedings of Machine Learning Research*, 7683–7694 (PMLR, 2020).
- [18] Cohen, J. M., Rosenfeld, E. & Kolter, J. Z. Certified Adversarial Robustness via Randomized Smoothing. *arXiv:1902.02918* (2019).
- [19] Noé, F., Olsson, S., Köhler, J. & Wu, H. Boltzmann generators: Sampling equilibrium states of many-body systems with deep learning. *Science* **365**, eaaw1147 (2019).
- [20] Schwalbe-Koda, D. & Gomez-Bombarelli, R. Supramolecular Recognition in Crystalline Nanocavities Through Monte Carlo and Voronoi Network Algorithms. *Journal of Physical Chemistry C* **125**, 3009–3017 (2021).
- [21] Schwalbe-Koda, D. & Gomez-Bombarelli, R. Benchmarking binding energy calculations for organic structure-directing agents in pure-silica zeolites. *Journal of Chemical Physics* **154**, 174109 (2021).
- [22] Smith, J. S., Isayev, O. & Roitberg, A. E. ANI-1: an extensible neural network potential with DFT accuracy at force field computational cost. *Chemical Science* **8**, 3192–3203 (2017).
- [23] Gao, X., Ramezanghorbani, F., Isayev, O., Smith, J. S. & Roitberg, A. E. TorchANI: A Free and Open Source PyTorch-Based Deep Learning Implementation of the ANI Neural Network Potentials. *Journal of Chemical Information and Modeling* **60**, 3408–3415 (2020).

REVIEWERS' COMMENTS

Reviewer #1

In my original comment, I questioned the efficiency of the proposed sampling approach in comparison to the traditional procedure of generating an MD trajectory once and then subsampling a training set from it (in a stratified way). Yet, in their response, the authors focus on an alternative scenario, where an active learning procedure based on some uncertainty measure is used. Of course, the latter approach is a more favorable baseline as it is significantly more expensive.

For small systems, it usually does not take many steps for an MD trajectory to converge and therefore generate a more or less ideal training set. This implies that no active learning scheme can significantly improve upon that training set. Since the authors strongly focus on rather small systems, I remain unconvinced that the proposed bootstrapping procedure can offer advantages from a strict computational complexity theory perspective.

With that being said, the authors rightfully point out that their contribution constitutes an algorithmic advance that circumvents the need for serial MD simulations in place of many shorter trajectories. This can indeed be of great benefit in practice.

With regard to my questioning the applicability of the proposed sampling approach to larger systems, I thank the authors for demonstrating its effectiveness in the additional zeolite-molecule example. I appreciate that the sampling problem becomes increasingly more difficult with system size (and therefore harder to capture with basic MD), which was the reason for my original skepticism.

All of my remaining objections have been adequately addressed by the authors in the revised manuscript. I thank the authors for taking the time to respond to my comments in such great detail.

Reviewer #2 (Remarks to the Author):

The authors have addressed my concerns and have considerably improved their manuscript. I am happy for the manuscript to be published.

Massachusetts Institute of Technology
77 Massachusetts Avenue
Cambridge, MA 02139

July 20, 2021

Reviewer 1

***Reviewer 1:** In my original comment, I questioned the efficiency of the proposed sampling approach in comparison the the traditional procedure of generating an MD trajectory once and then subsampling a training set from it (in a stratified way). Yet, in their response, the authors focus on an alternative scenario, were an active learning procedure based on some uncertainty measure is used. Of course, the latter approach is a more favorable baseline as it is significantly more expensive.*

For small systems, it usually does not take many steps for an MD trajectory to converge and therefore generate a more or less ideal training set. This implies that no active learning scheme can significantly improve upon that training set. Since the authors strongly focus on rather small systems, I remain unconvinced that the proposed bootstrapping procedure can offer advantages from a strict computational complexity theory perspective.

Authors: For the specific scenario suggested by the reviewer, we agree that molecular dynamics simulations are simpler to deploy than the proposed bootstrapping strategy. This is certainly the case of our ammonia molecule, whose dataset could have been generated using MD trajectories. Nevertheless, we believe this experiment can serve as a didactic demonstration of the method for a simple system. Instead of attempting to prove its lower deployment complexity with respect to MD simulations, we use the ammonia molecule as a low-dimensional system for which this exploration of the phase space can be visualized. We also agree with the reviewer that, for such small systems, active learning loops are not necessary, especially since the configuration space is captured very well by the initial data generation scheme. Thus, the goal of finding poorly explored regions of the phase space is indeed met by MD simulations for small systems, but may not be enough for larger systems, as the reviewer later points out.

Reviewer 1: With that being said, the authors rightfully point out that their contribution constitutes an algorithmic advance that circumvents the need for serial MD simulations in place of many shorter trajectories. This can indeed be of great benefit in practice.

With regard to my questioning the applicability of the proposed sampling approach to larger systems, I thank the authors for demonstrating its effectiveness in the additional zeolite-molecule example. I appreciate that the sampling problem becomes increasingly more difficult with system size (and therefore harder to capture with basic MD), which was the reason for my original skepticism. All of my remaining objections have been adequately addressed by the authors in the revised manuscript. I thank the authors for taking the time to respond to my comments in such great detail.

Authors: We thank the reviewer for the positive evaluation of our work and the interesting discussions.

Reviewer 2

Reviewer 2: The authors have addressed my concerns and have considerably improved their manuscript. I am happy for the manuscript to be published.

Authors: We thank the reviewer for the positive evaluation of our work and the insightful comments leading to this major improvement.